

# Morphofunctional changes in the immune system in colitis-associated colorectal cancer in tolerant and susceptible to hypoxia mice

Dzhuliia Dzhalilova[1], Maria Silina[1], Anna Kosyreva[1,2], Nikolai Fokichev[1,3] and Olga Makarova[1]

[1] Avtsyn Research Institute of Human Morphology, Petrovsky National Research Centre of Surgery, Moscow, Russia
[2] Research Institute of Molecular and Cellular Medicine, People's Friendship University of Russia (RUDN University), Moscow, Russia
[3] Faculty of Biology and Biotechnology, HSE University, Moscow, Russia

Corresponding author
Dzhuliia Dzhalilova,
juliajal93@mail.ru

## ABSTRACT

**Background**. One of the effective strategies for the treatment of tumor diseases, including colitis-associated colorectal cancer (CAC), is immunotherapy. During inflammation, NF-κB is activated, which is connected with the hypoxia-inducible factor—HIF, regulating the immune cells functioning and influences the CAC development. Organisms differ according to their hypoxia resistance and HIF expression. Therefore, the aim of the study was to characterize the thymus, spleen and mesenteric lymph nodes morphofunctional features, as well as changes in the subpopulation composition of peripheral blood cells and mesenteric lymph nodes in tolerant and susceptible to hypoxia C57Bl/6 mice in CAC.

**Methods**. Hypoxia tolerance was assessed by gasping time measurement in hypobaric decompression chamber. Based on the outcome, the mice were assigned to three groups characterized as 'tolerant to hypoxia', 'normal', and 'susceptible to hypoxia'. A month after determining hypoxia resistance CAC was modeled by intraperitoneal azoxymethane (AOM) administration and three cycles of dextran sulfate sodium consumption. Mice were sacrificed on the 141st day after the AOM administration, a morphological, morphometric and immunohistochemical study of tumors, morphological and morphometric study of thymus and spleen, and subpopulation composition of peripheral blood cells and mesenteric lymph nodes assessment were carried out.

**Results**. Tumors in tolerant and susceptible to hypoxia mice were represented by glandular intraepithelial neoplasia and adenocarcinomas, the area of which was larger in susceptible mice. Immunohistochemical study revealed a more pronounced Ki-67+ staining in tumors of susceptible mice. In CAC, only in tolerant mice, expansion of the thymic cortex was observed relative to the control group, while in susceptible ones, no changes were detected. Only in susceptible to hypoxia mice, spleen germinal centers of lymphoid follicles enlargement were observed. Only in susceptible mice during CAC, in comparison to the control group, the relative and absolute number of B-lymphocytes and relative—cytotoxic T-lymphocytes in blood increased. The relative cytotoxic T-lymphocytes and NK cells number in peripheral blood during CAC was higher in susceptible to hypoxia mice compared to tolerant ones. In susceptible to hypoxia mice,

more pronounced changes in the mesenteric lymph nodes subpopulation composition of cells were revealed—only in them the absolute and relative number of B-lymphocytes and NK cells, the absolute number of cytotoxic T-lymphocytes increased, and the relative number of macrophages decreased.

**Conclusions**. Morphofunctional differences in the thymus, spleen, mesenteric lymph nodes and blood immune cells reactions indicated the more pronounced immune response to the CAC development in susceptible to hypoxia mice, which should be taken into account in experimental studies.

# INTRODUCTION

Chronic inflammation can initiate tumor development (*Deng et al., 2016*; *Li, Zhou & Zhang, 2018*; *Niccolai et al., 2020*). It was demonstrated that approximately 25% of tumor cases were associated with chronic inflammation (*Balkwill & Mantovani, 2012*; *Murata, 2018*). According to the literature, this process contributes to the occurrence of repeated DNA damage in proliferating cells under the influence of proinflammatory cytokines, reactive oxygen and nitrogen species produced by leukocytes and other cells (*Lan, Chen & Wei, 2021*). This leads to the occurrence of point mutations, deletions or rearrangements in various genes, changes in the activity of the encoded proteins, constant activation of downstream signaling cascades and, as a consequence, to the excessive cell proliferation and tumor development (*Balkwill & Mantovani, 2012*; *Murata, 2018*). Gastric cancer and colorectal cancer (CRC) are common inflammatory-related malignancies (*Coussens & Werb, 2002*; *Grivennikov, Greten & Karin, 2010*). Chronic inflammation of the gastrointestinal tract, such as that occurring in gastritis and inflammatory bowel disease (IBD), significantly increases the tumor development risk through the production of proinflammatory cytokines such as TNFα, IL-1, IL-6, IL-17A and IL-23, and also includes activation of NF-κB and signal transducer and activator of transcription 3 (STAT3) (*West et al., 2015*; *Lasry, Zinger & Ben-Neriah, 2016*).

During early stages of tumor growth, activation of the immune system is observed (*West et al., 2015*). Numerous studies demonstrated that primary tumors tend to be densely infiltrated by various subpopulations of T-lymphocytes (*Galon et al., 2006*; *Camus et al., 2009*; *Pagès et al., 2009*; *Mlecnik et al., 2011*). T regulatory lymphocytes (Tregs) act as immunosuppressive cells, preventing excessive immune responses and maintaining immune tolerance (*Campbell & Koch, 2011*). Tumors were investigated to have high density of tumor-infiltrating Tregs (*Chaput et al., 2009*; *Ling et al., 2014*). Increased numbers of tumor-infiltrating Tregs are considered to promote tumor progression by mediating tumor-associated immunosuppression (*Campbell & Koch, 2011*). To develop effective immunotropic drugs for the CRC treatment it is critical to understand and predict the interactions between tumor and immune cells that lie in the balance of tumor development between immune surveillance and immune escape (*Neurath, 2019*).

The most adequate model of colitis-associated colorectal cancer (CAC) is the tumor development induction through the administration of azoxymethane (AOM) followed by the consumption of dextran sulfate sodium (DSS) by animals (*Neto et al., 2023*). However, the morphofunctional reactions of the immune system organs in this model are not sufficiently characterized. *Olguín et al. (2018)* on AOM-DSS model induced CAC demonstrated that in the early stages of its development there was a significant decrease in the number of Tregs in the blood and spleen, and in the later stages an increase in the content of Tregs in mesenteric lymph nodes (MLNs) was found.

Literature data on changes in the immune system organs in IBD are limited by data only about ulcerative colitis: in experimental acute ulcerative colitis induced by DSS, *Sasaki et al. (2008)* revealed a decrease in thymus weight and the increase in spleen weight in adult C57Bl/6 mice by 7 day of ulcerative colitis development. In addition, 5 days of 3% DSS consumption in C57Bl/6J mice was investigated to cause acute colon inflammation that persisted after replacing DSS with water (*Melgar, Karlsson & Michaëlsson, 2005*). *Fritsch Fredin et al. (2007)* demonstrated that in acute colitis caused by DSS, acute thymic involution was observed with devastation of the cortex and death of T-lymphocytes, but when inflammation becomes chronic, 3 weeks after the finishing of the DSS consumption cycle, thymus was being restored. According to *Postovalova et al. (2019a)* during a morphological study on the 28th and 56th days of the chronic ulcerative colitis development, a widening of the cortex in the thymus was observed in comparison to the control group and with the acute colitis group.

It is known that laboratory animals and humans could be divided according to their hypoxia tolerance, which is associated with differences in hypoxia-inducible factor-1 (HIF-1) expression, the activity of antioxidant enzymes, heat shock proteins, *etc.* (*Ghosh, Kumar & Pal, 2012*; *Kirova, Germanova & Lukyanova, 2013*; *Padhy et al., 2013*; *Jain et al., 2013*; *Jain et al., 2014*; *Mironova et al., 2019*; *Belosludtsev et al., 2020*). In addition, we previously demonstrated that the severity of diseases such as acute and chronic ulcerative colitis varies in mice with different hypoxia tolerance, which was largely determined by differences in the thymus and spleen reactions (*Dzhalilova et al., 2018a*; *Dzhalilova et al., 2018b*; *Dzhalilova, Tsvetkov & Makarova, 2024b*). Differences in HIF expression associated with hypoxia tolerance can obviously determine not only the severity of colitis, but also the initiation and rate of CAC progression, largely determined by the immune system reactions. It is known that HIF plays an important role in the development of IBD. It is activated in response to hypoxia, which develops in inflammation sites due to microcirculatory disorders and increased oxygen consumption by inflammatory cells (*Colgan, Furuta & Taylor, 2020*). Moreover, it was demonstrated that during tumor progression, HIF activation promotes immunosuppression, and its inhibition can lead to the antitumor immune activation (*McGettrick & O'Neill, 2020*). In T-lymphocytes, HIF-dependent induction of glycolysis is associated with the activation and differentiation of various T-cell subpopulations, in particular cytotoxic T-lymphocytes and memory cells (*McGettrick & O'Neill, 2020*). HIF-dependent changes in glycolysis promote differentiation of B-lymphocytes, high production of antibodies, as well as increased production of IL-10, which contributes immunosuppressive effect (*Cho et al., 2016*; *Li et al., 2021*; *Taylor & Scholz, 2022*). We

previously demonstrated (*Dzhalilova et al., 2024a*) during CAC that changes in the genes expression regulating hypoxia response, inflammation, cell cycle, apoptosis, and epithelial barrier function in tumors and the peritumoral area, as the number of CD3+CD8+ cytotoxic T-lymphocytes and vimentin+ cells, depend on the initial hypoxia tolerance. Higher expression of *Hif3a, Vegf, Tnfa, Il10, Tgfb, Cmet, Egf, Egfr, Egfr, Bax, Muc1*, and *Cldn7* was observed in the tumors of susceptible to hypoxia animals compared to tolerant mice, while only *Egf* was higher in the peritumoral area. Moreover, the content of F4/80+ macrophages, CD3-CD19+, CD3+CD4+ lymphocytes and NK cells in tumors did not differ between animals with different tolerance to hypoxia, but the number of CD3+CD8+ cytotoxic T lymphocytes and vimentin+ cells was higher in hypoxia susceptible mice. There is no data in literature characterizing morphofunctional features of the thymus, spleen and mesenteric lymph nodes in CAC with the regard to individual mice tolerance to hypoxia. Therefore, the purpose of the study was to characterize the morphofunctional changes in the immune system organs in mice tolerant and susceptible to hypoxia in AOM and DSS-induced CAC model.

## MATERIALS & METHODS

### Experimental animals

The study was made on male C57Bl/6 mice ($n = 60$) age 1.5–2 months, body weight 20–25 g, taken from the Branch of the Shemyakin-Ovchinnikov Institute of Bioorganic Chemistry of the Russian Academy of Sciences (BIBCh), Russia. Mice were housed ten per 42.5 × 27.6 × 15.3 cm cage with a footprint of 820 cm2 (Tecniplast, Italy), which are designed to hold up to 11 mice weighing 20–25 g or less, at regulated room temperature 25 °C $\pm$ 2 °C under 12:12 h light–dark cycle and 40–50% relative humidity with unlimited access to water and food (Char; JSC Range-Agro, Turakovo, Russia). All efforts were made to decrease suffering and possible stress for the animals and performed in accordance with the European Convention for the Protection of Vertebrate Animals Used for Experimental Use (ETS 123, Strasbourg, 1986), and Directives of the European Parliament and the Council of the European Union (2010/63/EU, Strasbourg, 2010). The study was approved by the Bioethics Committee at the Avtsyn Research Institute of Human Morphology (Protocol No. 36/12, March 28, 2022). All procedures were in accordance with the 'Animal Research Reporting of In Vivo Experiments' (ARRIVE) guidelines and the AVMA euthanasia guidelines 2020. According to literature, hypoxia tolerance depends on animal's sex. It was demonstrated that hypoxia tolerant organisms predominantly were detected among females, whereas susceptible and normal organisms predominantly among males, and there were sex differences in the morphofunctional state of the immune system (*Kosyreva et al., 2020*). To equalize sex differences, this study was performed only on male C57Bl/6 mice. Mice were randomly divided into the following experimental groups with a minimum five animals each. The study was conducted in two biological repeats. The first experiment was conducted in conjunction with a previously published study of gene expression changes in CAC (*Dzhalilova et al., 2024a*).

## Determination of hypoxia tolerance

The animal's tolerance to hypoxia was determined once in a mercury barometer-coupled decompression chamber, by simulated hypobaric hypoxia equivalent to 10,000 m altitude. Time till the first sign of characteristic hyperventilatory response ('gasping time') was recorded using the electronic stopwatch (*Kirova, Germanova & Lukyanova, 2013*; *Jain et al., 2013*; *Jain et al., 2014*; *Dzhalilova et al., 2018a*; *Pavlik et al., 2018*; *Mironova et al., 2019*). Based on the hypoxia tolerance determination results, mice were divided into 2 groups—tolerant ($n = 17$) with 'gasping time' was more than 10 min, and susceptible ($n = 18$)—gasping time was less than 3 min. Mice that were normally resistant ($n = 25$) to hypoxia ('gasping time' from 3 to 10 min) were not used in the experiments. Test time was between 08.30 am to 12.00 pm and testing order was randomized daily. For each animal, different researchers were involved as follows: a first researcher (DD) determined the hypoxia tolerance and was responsible for the cervical dislocation. This researcher was the only person aware of the group allocation. The second and the third researchers (MS, AM) performed the flow cytometry, morphological and morphometric study.

## Colitis-associated colorectal cancer model

For CAC modeling single AOM (A5486-25MG, Sigma Azoxymethan) intraperitoneal injection at a dose of 10 mg/kg (*Deng et al., 2022*) was performed a month after hypoxia resistance test among tolerant ($n = 12$) and susceptible ($n = 13$) to hypoxia mice. A week after the AOM injection, drinking water was replaced with 1% DSS (Dextran sulfate sodium salt, $M_r \sim 40,000$, AppliChem, Germany) for 7 days. After 14 days of water drinking, it was again replaced with 0.5% DSS for 7 days, then after 14 days of water drinking—for 5 days with 0.5% DSS (*Tanimura et al., 2021*). Clinical symptoms were assessed by the presence of diarrhea and blood in the feces of mice. Clinical manifestations occurred after each DSS consumption cycle, and then the animal's condition returned to normal. If throughout the experiment the mice tended to have a pronounced pain posture or/and continuous diarrhea and blood in the feces, and in cases when the tumor size reaches a maximum dimension of two cm in mice then this was the criteria established for euthanizing animals prior to the planned end. However, during the described investigation, there were no mice that were euthanized prior to the planned end of the experiment. No animal death was observed during the experiment. Cervical dislocation was used for the euthanizing from the experiment by 86 days after the last drinking water replacement with DSS, that is, on the 141st day of the experiment. Physiological solution (0.9% NaCl; Armavirskaya Biofabrika, Russia) was injected intraperitoneally in control groups of tolerant ($n = 5$) and susceptible ($n = 5$) to hypoxia mice, then mice consumed drinking water for 140 days, then they were euthanized from the experiment. There were no mice survived after 140 days of the experiment.

## Morphological and morphometric study

Data were collected as previously described in *Dzhalilova et al. (2024a)*. Specifically, colon was divided into proximal, medial and distal parts, buffered 10% formalin was used for fixation during 24 h. The tumor size did not reach a maximum dimension of 20 mm

in mice. Then colon parts were put in paraffin, stepped-serial histological sections were prepared (from eight to 12 sections) with a 200 μm step, and with hematoxylin and eosin staining. In the distal colon tumors area assessment was made on stepped-serial histological sections, in which tumors were detected. They were photographed at a magnification 50x using an Axioplan 2 Imaging microscope (Carl Zeiss, Oberkochen, Germany). The tumor borders outlining was performed with Adobe Photoshop CS6. Eventually, isolated tumors area was revealed with the Image-Pro Plus 6.0 program, and for each mouse their area was calculated.

A morphological study of the liver and lungs was performed on sections stained with hematoxylin and eosin to assess the presence of metastases. To assess the response of the immune system organs, a morphological study of the thymus, spleen and MLNs was performed. The mice lungs were fixed in Carnoy's solution (60 ml ethanol, 30 ml chloroform, and 10 ml glacial acetic acid) for 2 h, the liver, thymus and spleen were fixed in Bouin's solution (75 ml picric acid, 25 ml formalin, and five ml glacial acetic acid) and MLNs were fixed in buffered 10% formalin for 24 h, and organs were embedded in paraffin according to routine procedures. Histological sections of 4–5 μm thickness were produced and stained with hematoxylin and eosin (BioVitrum). In the histological slices of the thymus and spleen, the volume fraction of the functional zones was determined by the point-count method.

## Immunohistochemical study

Immunohistochemical detection of Ki-67 was carried out by sandwich method. Colon sections were deparaffinized, citrate buffer pH 6.0 with 0.5% Tween-20 at 100 °C was used for the antigen retrieval and blocked in phosphate-buffered saline with 0.1% bovine serum albumin at room temperature before exposure to antibodies (primary: Rabbit Monoclonal Anti-Ki-67, ab16667, Abcam, Cambridge, UK; secondary: Goat Anti-Rabbit IgG H&L Alexa Fluor 488 ab150077, Abcam). After PBS cleansing, DAB work solution was amplified until color changing visualized. Digital images were photographed using an Axioplan 2 Imaging microscope (Carl Zeiss).

## Flow cytometry

Blood was taken from the jugular veins, and the absolute and relative number of lymphocytes was determined using the automatic hematology analyzer Mindray BC-2800Vet (China). Via flow cytometry with a Cytomics FC 500 (Beckman Coulter), the subpopulation composition of peripheral blood cells and MLNs was determined in mice of the control and experimental groups. Peripheral blood from the jugular veins was collected into tubes containing EDTA as an anticoagulant. Isolation of cells from MLNs was performed with Potter homogenizer (*Winter et al., 2019*). To analyze the cells, antibodies conjugated with FITC (fluorescein isothiocyanate), PE (phycoerythrin), PE-Cy5 and PE-Cy7, to CD19+ (B-lymphocytes), CD3e+CD4+ (T-helper cells), CD3e+CD8+ (cytotoxic T-lymphocytes), CD4+CD25+Foxp3+ (regulatory T-lymphocytes), NK-1.1 (NK cells), CD11b+ (monocytes), F4/80 (macrophages) from eBioscience were used. Lymphocytes and monocytes were selected based on their morphology using forward-versus side-scatter

(FSC-SSC) dotplots because the samples were not fixed, erythrocytes were lysed, no more than 4–6 h passed after obtaining the materials and before the study on the flow cytometer, the cells were constantly at room temperature. Combining anti-CD4 and anti-CD8, we identified CD3e+CD8+ and CD3e+CD4+ T cells, regulatory T-lymphocytes were identified on CD4+ T cells by CD25+Foxp3+ expression, monocytes were gated by CD11b+ expression. For flow cytometry data analysis FlowJo software was used (Becton, Dickinson & Company, USA).

## Statistical analysis

The data obtained were statistically processed with GraphPad Prism 8.0. Kolmogorov–Smirnov criterion was used for the type of the indicators distribution determination. In case the data were not distributed normally, the differences between indicators significance were determined *via* the nonparametric Mann–Whitney, Kruskal-Wallis, and Dunn tests. Data were represented as median and interquartile range Me (LQ(25%); UQ(75%)). Fisher's test was used to assess the statistical significance of clinical symptoms. Differences were considered statistically significant at $p < 0.05$.

# RESULTS

## CAC development frequency and tumor morphological features

During clinical observation of mice that received a single dose of AOM and three cycles of DSS, both tolerant and susceptible to hypoxia mice experienced diarrhea and the blood detected in the feces, which was associated with the development of ulcerative colitis and CAC (Fig. 1A). No differences were found when analyzing the statistical significance of the symptoms by Fisher's test ($p = 0.3$).

During morphological study in the distal colon tumors were detected in 41.6% of the tolerant (five of 12) and in 84.6% of the susceptible (11 of 13) to hypoxia mice, in the medial colon—in 8.3% of the tolerant (one of 12) and in 7.7% of susceptible (one out of 13), no tumors were detected in the proximal colon of both tolerant and susceptible mice. In the distal colon, tumors were represented by glandular intraepithelial neoplasia (80% of all detected tumors in tolerant, 0% in susceptible), as well as adenocarcinomas (20% of all detected tumors in tolerant, 100% in susceptible), their frequency development was higher ($p = 0.003$) in susceptible mice (Fig. 1A). Colon histological examination in both tolerant and susceptible to hypoxia mice revealed chronic colitis: single epithelized ulcers were detected in the mucous membrane, the crypts were deformed, their lumens were enlarged, and crypt abscesses were noted. Inflammatory infiltration of lymphocytes, macrophages and plasma cells was observed, mostly pronounced in the basal part of the mucous membrane lamina propria. Glandular intraepithelial neoplasia was characterized by clusters of several crypts lined with hyperchromatic polymorphic epithelial cells with dilated and deformed lumens (Fig. 1B). Exophytic growth of adenocarcinomas was observed; microscopically, adenocarcinomas were represented by many glands lined with proliferating atypical columnar epithelium; crypt abscesses were detected in some crypts - lumens of the crypts were dilated and filled with mucus with a large neutrophils number. The tumor's stroma was represented by connective tissue with partially ordered thin

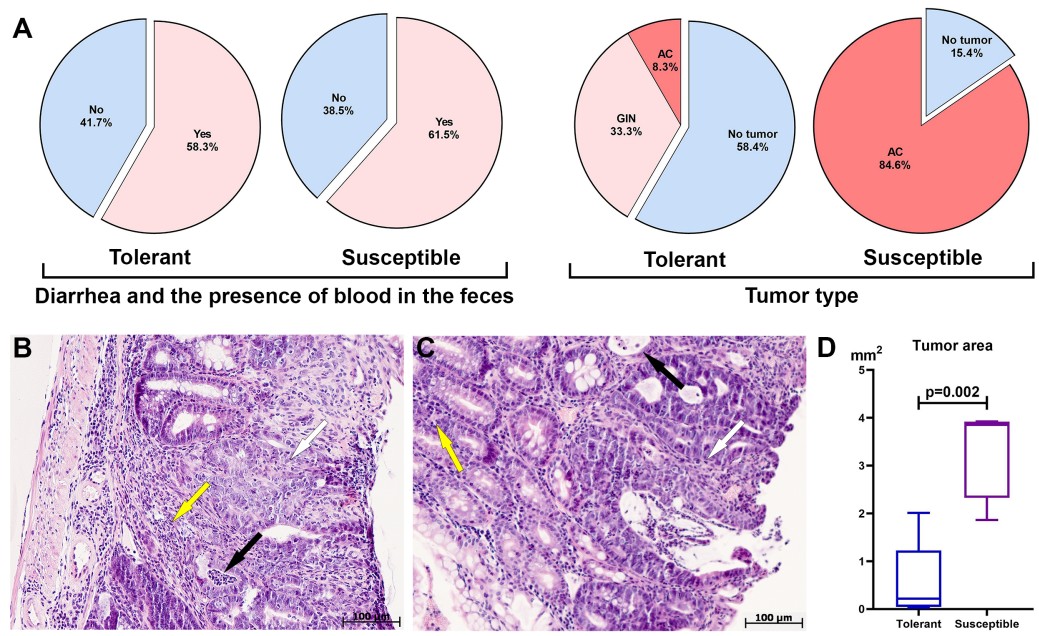

**Figure 1 Clinical manifestations, distal colon tumors incidence and morphological characteristics on the 141st day after AOM administration.** (A) CAC clinical manifestations, tumor type and development frequency in the distal colon in tolerant and susceptible to hypoxia mice, GIN, glandular intraepithelial neoplasia; AC, adenocarcinoma; (B) glandular intraepithelial neoplasia (tolerant to hypoxia mice), the black arrow indicates crypt abscesses, yellow–inflammatory infiltrate, white–disruption of crypt architectonics; (C) adenocarcinoma (susceptible to hypoxia mice), the black arrow indicates crypt abscesses, yellow–inflammatory infiltrate, white–disruption of crypt architectonics; (D) tumor area in tolerant ($n = 5$) and susceptible ($n = 11$) to hypoxia mice, Me (25–75%). Note: $p$–statistical significance of differences, Mann–Whitney test.

fibers; diffuse connective tissue infiltration was noted with a small number of lymphocytes, macrophages and focal neutrophils (Fig. 1C).

According to the morphometric study, the tumor area in the distal colon in susceptible to hypoxia mice was statistically significantly higher ($p = 0.002$) in comparison to tolerant (Fig. 1D). Morphological study of the liver, lungs and MLNs of both tolerant and susceptible to hypoxia mice did not reveal metastases. For results of morphometric immune system organs study and flow cytometry analysis, were used data obtained only from mice with confirmed tumor, *i.e.,* with glandular intraepithelial neoplasia and adenocarcinomas of tolerant ($n = 5$) and susceptible ($n = 11$) to hypoxia mice.

## Tumor immunohistochemical study

During all cell cycle phases except G0 the Ki-67 antigen is expressed and therefore it is commonly used as proliferation marker (*Gerdes et al., 1984*). In control groups of tolerant and susceptible to hypoxia mice there were single Ki-67+ cells in the colon mucosa. During CAC, glandular intraepithelial neoplasia and adenocarcinoma both demonstrated an increase in the expression of Ki-67 staining compared with the control groups. More pronounced Ki-67+ staining in tumors were in adenocarcinomas in susceptible to hypoxia mice (Fig. S1).

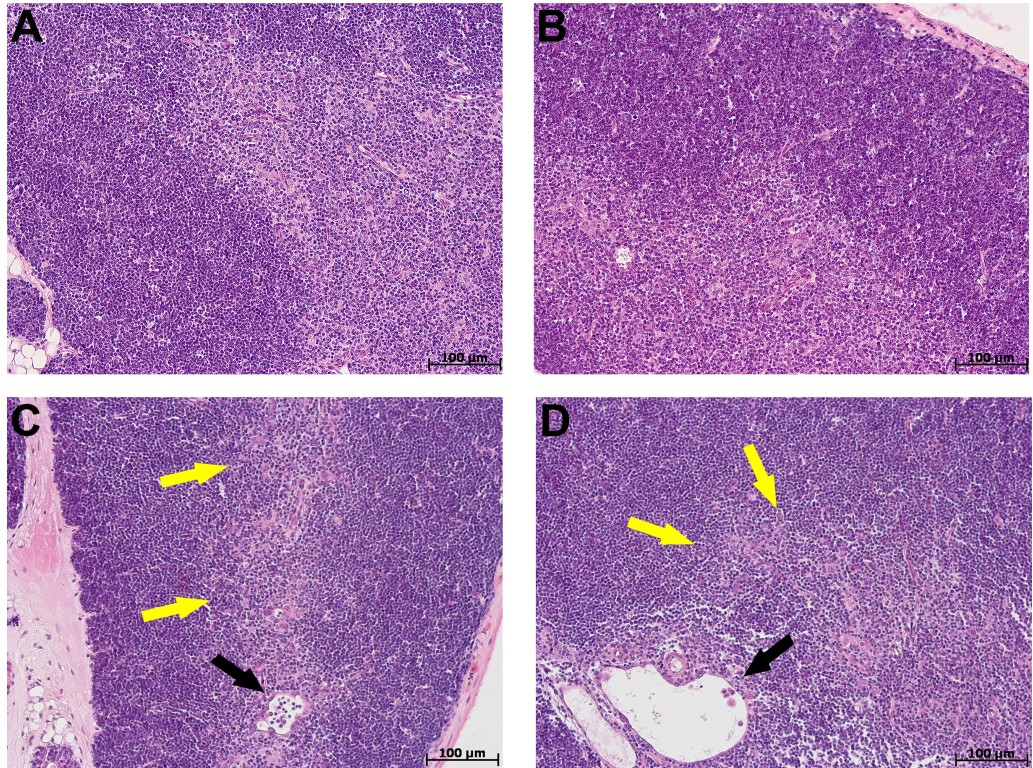

**Figure 2** **Thymus morphological study in tolerant ($n_{control}$ = 5, $n_{cac}$ = 5) and susceptible ($n_{control}$ = 5, $n_{cac}$ = 11) to hypoxia mice of the control (A, B) and experimental (C, D) groups.** (A, B) Control groups, cortex is densely populated with lymphocytes, and the border between the cortex and medulla is clear; (C) experimental group, black arrow indicates thymic body in the form of cyst-like cavity, yellow–focally unclear boundaries between the cortex and medulla; (D) experimental group, black arrow indicates thymic body in the form of cyst-like cavity, yellow–unclear boundaries between the cortex and medulla. (A, C) Tolerant to hypoxia; (B, D) susceptible to hypoxia. Hematoxylin and eosin staining.

## Thymus, spleen and MLNs morphological and morphometric study during CAC

At morphological examination in the thymus of control group mice, the cortex was represented by densely arranged lymphocytes and epithelial cells (Figs. 2A, 2B). The boundaries between the cortex and medulla were clear. The thymus subcapsular zone was represented by 4–8 rows of lymphoblasts. The thymus medulla was represented by evenly distributed epithelial cells and lymphocytes. Thymic bodies in the form of cyst-like cavities, as well as those consisting of 3–5 epithelial cells and single ones with small keratohyalin deposits were detected in it.

The morphological study in the thymus of tolerant to hypoxia mice during CAC in comparison to the control group revealed the cortex widening, the boundaries between the cortex and medulla were focally unclear. Thymic bodies consisted of 3–5 cells, more than five cells, numerous thymic bodies with keratohyalin and in the form of cyst-like cavities were found (Fig. 2C). In susceptible to hypoxia mice, focal devastation of the cortex and focally unclear boundaries between the cortex and medulla were revealed. Thymic bodies

consisted of 3–5 cells, more than five cells, a few thymic bodies with keratohyalin and in the form of cyst-like cavities were detected (Fig. 2D).

The thymus morphometric study in mice tolerant and susceptible to hypoxia in the control groups did not reveal any differences. During CAC, cortex widening and medulla narrowing were observed in tolerant to hypoxia mice in comparison to the control group, while in the susceptible to hypoxia mice no differences were detected. In tolerant to hypoxia mice of the experimental group in comparison to the susceptible, the volume fraction of the cortex was higher, and the medulla was lower (Fig. 3). When studying thymic bodies of various types, an increase in the number of thymic bodies in the form of cyst-like cavities was found only in tolerant to hypoxia mice on the 141st day of the experiment in comparison to the control group (Table 1).

In the spleen morphological study during CAC, lymphoid follicles with and without small germinal centers were detected in both tolerant and susceptible to hypoxia mice; single mitoses were detected in them. The PALS (the periarterial lymphatic sheaths) zone was moderately and weakly expressed. The number of megakaryocytes was increased (Fig. 4).

In the spleen morphometric analysis in tolerant to hypoxia control group mice, the lymphoid follicles volume fraction was larger, and the PALS zones were smaller compared to susceptible. During CAC, in both the tolerant and susceptible to hypoxia mice, an increase in the lymphoid follicles volume fraction and a decrease in the proportion of the PALS zone relative to the white pulp were demonstrated, while the lymphoid follicles germinal centers expansion was observed only in susceptible to hypoxia mice (Fig. 5).

The morphological study of MLNs in the control groups of tolerant and susceptible mice did not reveal any differences, lymphoid follicles with and without germinal centers; in the germinal centers, cells were loosely located, represented by lymphoblasts, lymphocytes, macrophages, and single dying cells. The paracortical zone is focally devastated, the marginal sinuses were not dilated, they contained lymphocytes and macrophages. Intermediate sinuses were not pronounced. In the cerebral sinus the ratio of lymphocytes and macrophages was 1:1. The pulp strands stroma was populated by densely located mature lymphocytes and macrophages (Fig. 6).

In the morphological study of MLNs during CAC, in both tolerant and susceptible to hypoxia mice, the outer cortex and paracortical zone were expanded and densely populated with lymphocytes. Lymphoid follicles without and with germinal centers were revealed. There were loose and densely located lymphoblasts and lymphocytes, fragments of nuclei in the germinal centers. A large number of plasma cells, as well as lymphocytes and macrophages, were detected in the medullary cords. The marginal and medullary sinuses were dilated, there were many lymphocytes and macrophages in them. Macrophages with granular eosinophilic contents, single foreign-body giant cell were detected. Thus, in both tolerant and susceptible to hypoxia adult male C57Bl/6 mice on the 141st day after AOM administration, according to a morphological study of MLNs, B- and T-zones hyperplasia were revealed. A large number of plasma cells were detected in the medullary cords.

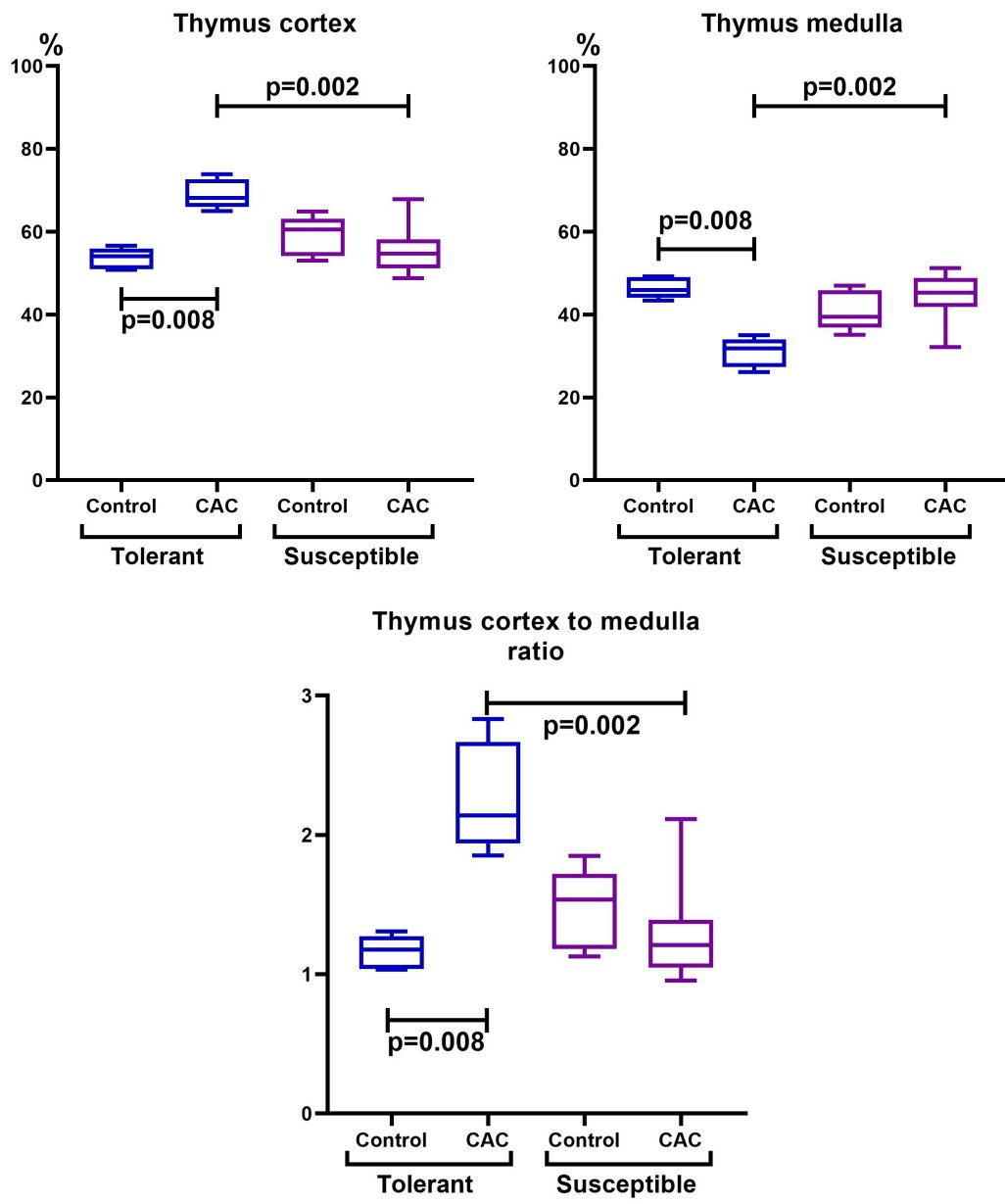

**Figure 3** Cortex and medulla thymic volume fractions and their ratio in tolerant ($n_{control}$ = 5, $n_{cac}$ = 5) and susceptible ($n_{control}$ = 5, $n_{cac}$ = 11) to hypoxia mice of the control and experimental groups. Me (25–75%). Note: $p$–statistical significance of differences, Mann–Whitney, Kruskal–Wallis and Dunn tests.

## Blood and MLNs during CAC flow cytometry

In the control group, the absolute and relative number of lymphocytes was higher in susceptible to hypoxia mice in comparison with tolerant. On the 141st day of the experiment, only in susceptible to hypoxia mice there was a statistically significant decrease in the relative content of lymphocytes (Table 2). It was demonstrated that in the control group that the absolute number of NK cells and CD11b+ monocytes in the blood was

**Table 1 Different types thymic bodies content in tolerant and susceptible to hypoxia mice in CAC.** Me (25–75%). *p*–statistical significance of differences, Mann–Whitney, Kruskal–Wallis and Dunn tests.

| | Control | | CAC | | *p* |
|---|---|---|---|---|---|
| | Tolerant[1] | Susceptible[2] | Tolerant[3] | Susceptible[4] | |
| Thymic bodies in the form of 3–5 cell clusters, % | 57.89 (42.07–70.00) | 50.84 (43.99–58.80) | 41.54 (29.63–52.66) | 60.48 (52.63–68.42) | $p^{1,2} = 0.38$ $p^{3,4} = 0.06$ $p^{1,3} = 0.21$ $p^{2,4} = 0.18$ |
| Thymic bodies in the form of clusters containing more than 5 cells, % | 10.00 (2.78–29.88) | 13.31 (9.99–16.85) | 7.38 (5.10–8.96) | 13.76 (8.33–21.14) | $p^{1,2} = 0.88$ $p^{3,4} = 0.07$ $p^{1,3} = 0.55$ $p^{2,4} = 1.00$ |
| Thymic bodies with keratinization, % | 16.67 (10.27–23.18) | 15.00 (9.00–19.34) | 10.00 (7.82–11.39) | 15.23 (8.64–16.67) | $p^{1,2} = 0.45$ $p^{3,4} = 0.17$ $p^{1,3} = 0.11$ $p^{2,4} = 0.98$ |
| Thymic bodies in the form of cyst-like cavities, % | 10.00 (8.19–16.37) | 18.00 (13.72–29.76) | 42.86 (31.95–51.62) | 5.15 (1.36–31.47) | $p^{1,2} = 0.15$ $\boldsymbol{p^{3,4} = 0.003}$ $\boldsymbol{p^{1,3} = 0.008}$ $p^{2,4} = 0.14$ |

**Notes.**
*Numbers on values refer to each animal group.
*$p$ values less than 0.05 are highlighted in bold.

higher in susceptible to hypoxia in comparison with tolerant mice. In the experimental groups of both tolerant and susceptible to hypoxia mice, the absolute number of NK cells and T-regulatory lymphocytes and the relative number of CD3e+CD4+ T-lymphocytes in the blood decreased in comparison to the control groups, while the relative number of CD11b+ monocytes increased. However, only in susceptible to hypoxia mice the relative and absolute numbers of CD19+ and relative - CD3e+CD8+ lymphocytes in blood increased. The relative number of CD3e+CD8+ and NK cells in the peripheral blood during CAC was higher in susceptible to hypoxia in comparison to tolerant mice.

In the control group of susceptible to hypoxia mice in the MLNs, the absolute CD3e+CD4+ T-lymphocytes number was lower compared to tolerant. At the same time, on the 141st day of the experiment, in comparison with the control group, no changes were observed in the mice tolerant to hypoxia, and in the susceptible to hypoxia mice the relative and absolute CD19+ and NK cells number, the absolute CD3e+CD8+ lymphocytes number increased, and F4/80+ macrophages relative number decreased (Figs. 7 and 8). The relative and absolute numbers of CD19+ and NK cells in MLNs in CAC were higher in susceptible mice in comparison to tolerant (Figs. 7 and 8).

The summary results are presented in Figs. 9. The identified differences indicated the more pronounced reaction of immune cells to the CAC development in susceptible to hypoxia mice at both the local and systemic levels.

## DISCUSSION

The morphofunctional state of the immune system in tolerant and susceptible to hypoxia mice in control group differed. In susceptible to hypoxia mice in comparison to tolerant,

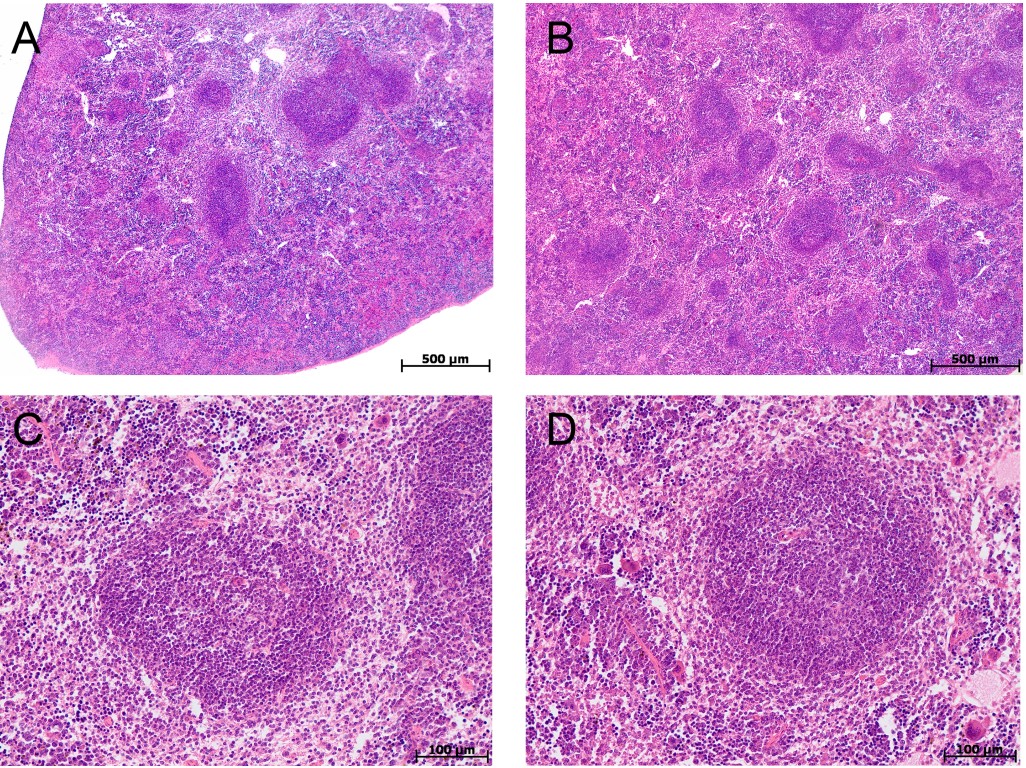

**Figure 4** **Spleen morphological study in tolerant and susceptible to hypoxia mice on the 141st day of the experiment.** (A, C) Tolerant to hypoxia ($n = 5$), (B, D) susceptible to hypoxia ($n = 11$). (A, B) Spleen white pulp, the PALS zone was moderately and weakly expressed, lymphoid follicles with and without germinal centers. (C, D) Lymphoid follicles with germinal centers. Hematoxylin and eosin staining.

a larger PALS volume fraction and a smaller lymphoid follicles volume fraction in the spleen white pulp, higher absolute number of lymphocytes, NK cells and monocytes in the peripheral blood, as well as a lower T-helpers number in MLNs were revealed. The obtained data may be caused by hypoxic effects in a decompression chamber during tolerance to the lack of oxygen determination.

CAC development and progression in susceptible to hypoxia mice was characterized by a faster rate. On the 141st day of the experiment, in susceptible to hypoxia mice, there was a higher tumor development incidence in comparison to tolerant ones. In addition, in susceptible mice, 100% of the detected tumors were adenocarcinomas, while in tolerant— 20% adenocarcinomas and 80% glandular intraepithelial neoplasia. Morphometric analysis demonstrated that the tumor areas were significantly higher in susceptible to hypoxia mice. Immunohistochemical study also revealed a more pronounced Ki-67+ staining in tumors of susceptible mice, indicating a high tumor progression rate.

Most likely, these differences were largely determined by the immune system functioning. Impaired functioning of immune cells was considered an important factor contributing to the CRC development. During the CRC initiation and progression, a significant decrease

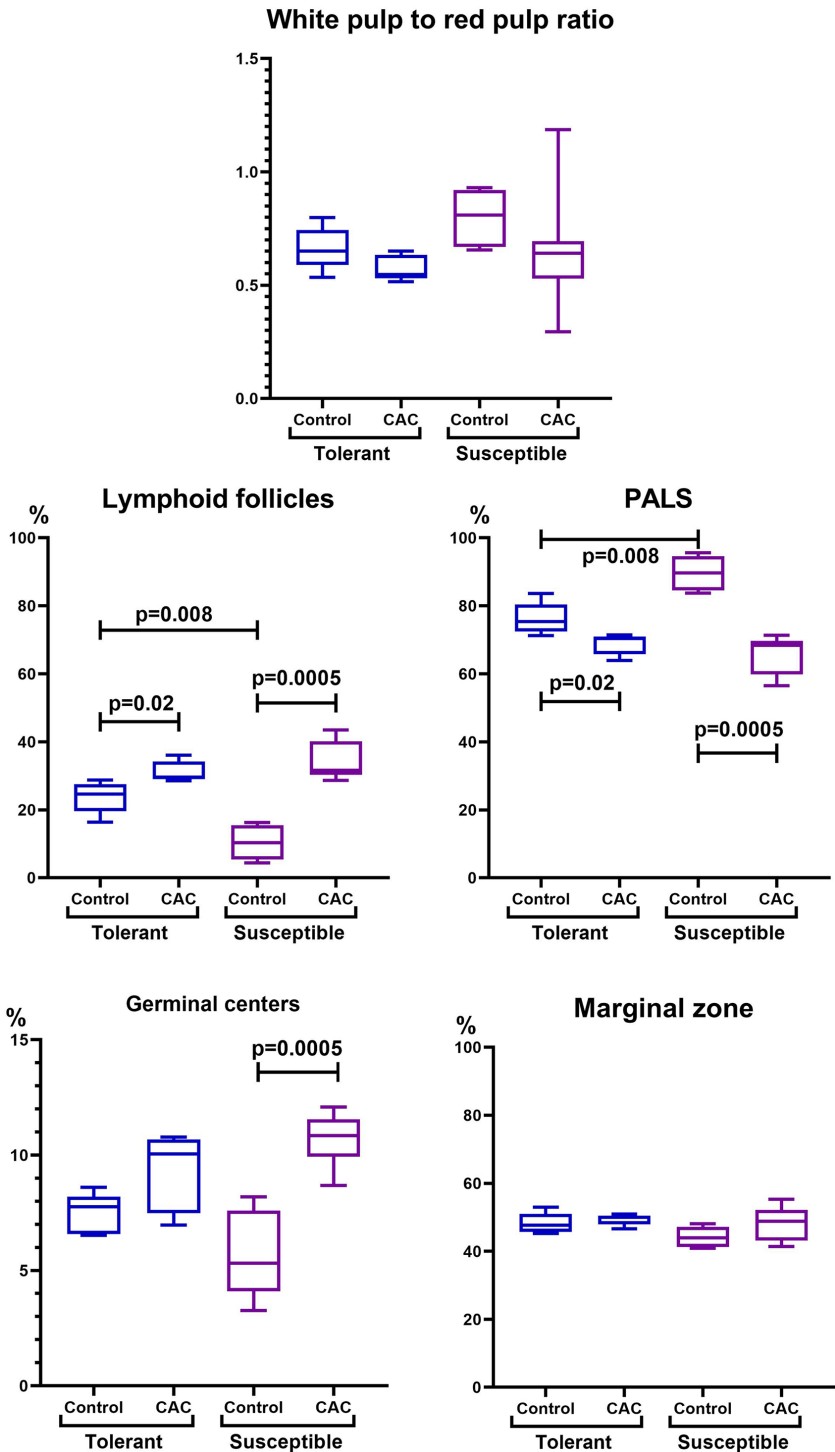

**Figure 5** **Spleen morphofunctional features in tolerant (n$_{control}$ = 5, n$_{cac}$ = 5) and susceptible (n$_{control}$ = 5, n$_{cac}$ = 11) to hypoxia in mice of the control and experimental groups.** Ratio of the spleen white to red pulp volume fraction, lymphoid follicles and the PALS zone volume fraction, and the lymphoid follicles germinal centers and marginal zones volume fraction in the tolerant and susceptible to hypoxia spleen. Me (25–75%). Note: *p*–statistical significance of differences, Mann–Whitney, Kruskal–Wallis and Dunn tests.

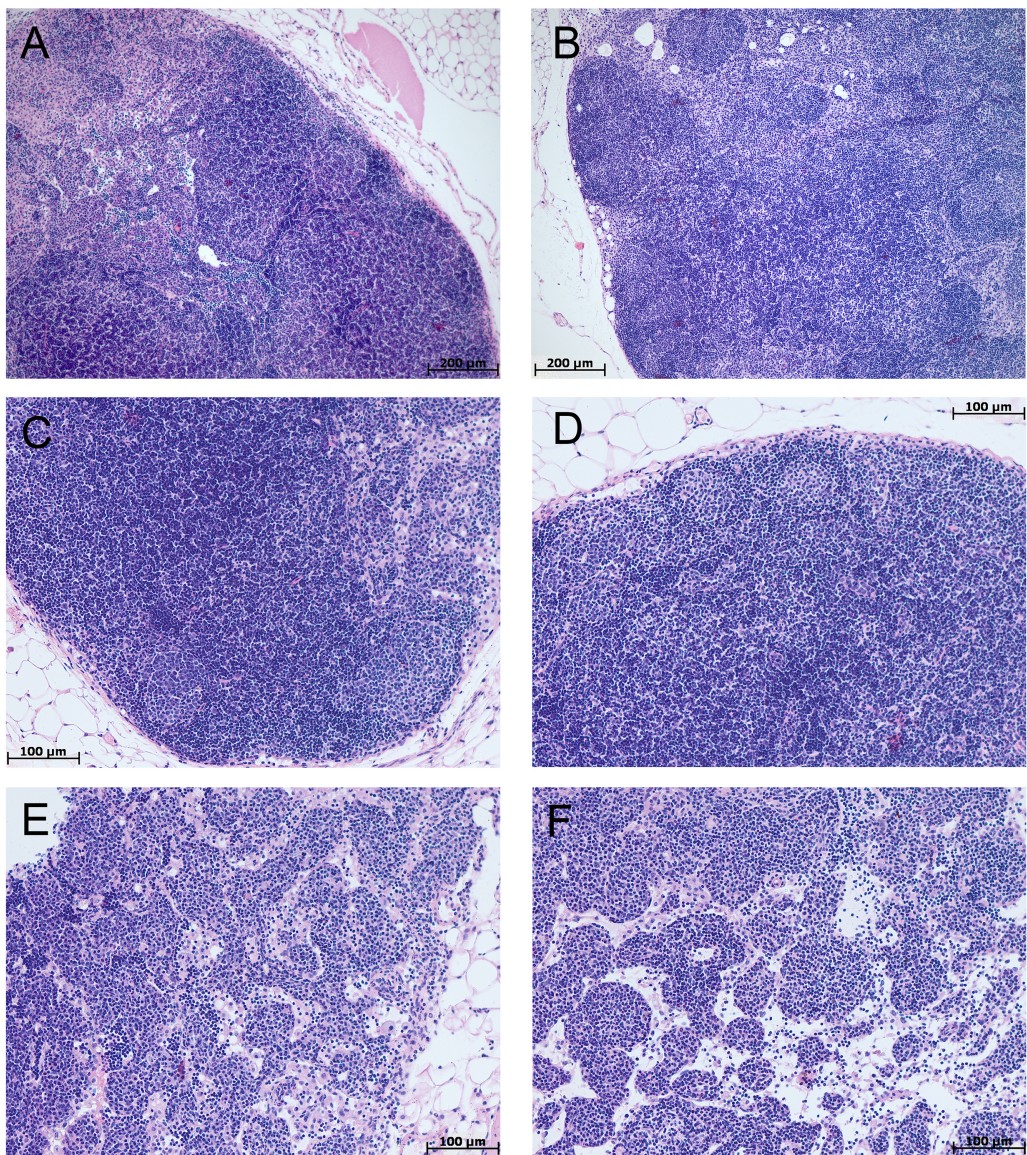

**Figure 6** **MLNs morphological study in tolerant ($n_{control} = 5$, $n_{cac} = 5$) and susceptible ($n_{control} = 5$, $n_{cac} = 11$) to hypoxia mice of the control (A, B) and experimental (C–F) groups.** (A, B) Control groups, lymphoid follicles with and without germinal centers; in the germinal centers, the cells were loosely located, represented by lymphoblasts, lymphocytes, macrophages, and single dying cells. (C–F) Experimental groups, the outer cortex and paracortical zone were expanded and densely populated with lymphocytes. Lymphoid follicles without and with germinal centers. The marginal and medullary sinuses were dilated, there were many lymphocytes in them. (A, C, E) Tolerant to hypoxia, (B, D, F) susceptible to hypoxia. Hematoxylin and eosin staining.

in the organism's antitumor immunity is observed, mainly due to the tumor cells escape from the immune system (*Croci et al., 2007*; *McLean et al., 2011*; *Shi et al., 2015*).

According to literature, thymus morphological changes in chronic ulcerative colitis are manifested by cortex widening, more pronounced in the later stages of the disease, and

**Table 2  Flow cytometry in peripheral blood.** Absolute and relative numbers of lymphocytes, CD3e+CD4+, CD3e+CD8+, CD4+CD25+Foxp3+, NK1.1+, CD19+, CD11b+ cells in the peripheral blood of tolerant and susceptible to hypoxia mice in control groups and in CAC. Me (25–75%). Note: $p$–statistical significance of differences, Mann–Whitney, Kruskal–Wallis and Dunn tests.

| | | Control | | CAC | | $p$ |
|---|---|---|---|---|---|---|
| | | Tolerant[1] | Susceptible[2] | Tolerant[3] | Susceptible[4] | |
| Lymphocytes | $\times 10^6$/ml | 4.40 (4.00–6.35) | 12.30 (7.20–13.10) | 4.50 (3.00–8.85) | 7.10 (6.80–8.00) | $\boldsymbol{p^{1,2} = 0.03}$ $p^{3,4} = 0.58$ $p^{1,3} = 0.59$ $p^{2,4} = 0.07$ |
| | % | 66.50 (62.80–68.35) | 74.90 (73.25–77.75) | 54.30 (49.80–65.80) | 65.10 (63.30–70.20) | $\boldsymbol{p^{1,2} = 0.008}$ $p^{3,4} = 0.09$ $p^{1,3} = 0.09$ $\boldsymbol{p^{2,4} = 0.0005}$ |
| T-helper cells (CD3e+CD4+) | $\times 10^6$/ml | 2.191 (1.558–3.425) | 5.749 (3.131–6.719) | 0.563 (0.173–2.062) | 1.291 (1.123–1.413) | $p^{1,2} = 0.1$ $p^{3,4} = 0.58$ $p^{1,3} = 0.1$ $\boldsymbol{p^{2,4} = 0.0005}$ |
| | % | 45.90 (34.75–56.25) | 41.70 (32.15–52.70) | 12.80 (6.38–23.20) | 17.10 (15.80–19.40) | $p^{1,2} = 0.31$ $p^{3,4} = 0.44$ $\boldsymbol{p^{1,3} = 0.02}$ $\boldsymbol{p^{2,4} = 0.0005}$ |
| Cytotoxic T-lymphocytes (CD3e+CD8+) | $\times 10^6$/ml | 1,398 (0.162–1.628) | 1.306 (1.222–1.704) | 0.359 (0.219–1.054) | 1.548 (1.428–1,936) | $p^{1,2} = 1.00$ $\boldsymbol{p^{3,4} = 0.001}$ $p^{1,3} = 0.55$ $p^{2,4} = 0.09$ |
| | % | 14.06 (4.86-18.45) | 10.30 (9.59-13.42) | 8.15 (6.91-11.90) | 24.30 (20.50-28.70) | $p^{1,2} = 0.84$ $\boldsymbol{p^{3,4} = 0.001}$ $p^{1,3} = 0.31$ $\boldsymbol{p^{2,4} = 0.0005}$ |
| Regulatory T-cells (CD4+CD25+Foxp3+) | $\times 10^6$/ml | 0.0417 (0.0237–0.0695) | 0.0599 (0.0408–0.0611) | 0.0098 (0.0004–0.0011) | 0.0010 (0.0006–0.0029) | $p^{1,2} = 0.55$ $p^{3,4} = 0.51$ $\boldsymbol{p^{1,3} = 0.008}$ $\boldsymbol{p^{2,4} = 0.0005}$ |
| | % | 36.50 (29.35–42.00) | 37.00 (28.13–39.75) | 1.40 (0.68–10.57) | 25.30 (3.71–29.50) | $p^{1,2} = 1.00$ $p^{3,4} = 0.06$ $\boldsymbol{p^{1,3} = 0.008}$ $p^{2,4}=0.06$ |
| NK (NK1.1+) | $\times 10^6$/ml | 2.18 (1.50–2.22) | 3.60 (3.35–4.04) | 0.20 (0.19–0.65) | 0.61 (0.46–1.26) | $\boldsymbol{p^{1,2} = 0.008}$ $p^{3,4} = 0.07$ $\boldsymbol{p^{1,3} = 0.008}$ $\boldsymbol{p^{2,4} = 0.0005}$ |
| | % | 21.20 (20.20–27.90) | 21.80 (17.10–24.11) | 2.53 (2.49–3.01) | 10.03 (5.93–20.85) | $p^{1,2} = 0.69$ $\boldsymbol{p^{3,4} = 0.008}$ $\boldsymbol{p^{1,3} = 0.0005}$ $p^{2,4}=0.06$ |

| | | Control | | CAC | | $p$ |
|---|---|---|---|---|---|---|
| | | Tolerant[1] | Susceptible[2] | Tolerant[3] | Susceptible[4] | |
| B-lymphocytes (CD19+) | $\times 10^6$/ml | 0.562 (0.234–1.004) | 0.964 (0.792–1.143) | 0.713 (0.275–2.314) | 1.875 (1.677–2.060) | $p^{1,2} = 0.15$ $p^{3,4} = 0.22$ $p^{1,3} = 0.69$ $\boldsymbol{p^{2,4} = 0.0005}$ |
| | % | 10.23 (6.53–13.00) | 7.53 (6.74–9.24) | 16.20 (8.59–26.35) | 29.85 (23.00–30.87) | $p^{1,2} = 0.55$ $p^{3,4} = 0.06$ $p^{1,3} = 0.22$ $\boldsymbol{p^{2,4} = 0.0005}$ |
| Monocytes (CD11b+) | $\times 10^6$/ml | 0.250 (0.148–0.259) | 0.330 (0.312–0.451) | 2.100 (1.144–4.507) | 0.986 (0.401–1.787) | $\boldsymbol{p^{1,2} = 0.008}$ $p^{3,4} = 0.15$ $\boldsymbol{p^{1,3} = 0.008}$ $p^{2,4} = 0.07$ |
| | % | 2.41 (2.11–3.29) | 1.94 (1.83–2.26) | 24.90 (13.71–27.15) | 18.88 (12.84–20.90) | $p^{1,2} = 0.22$ $p^{3,4} = 0.07$ $\boldsymbol{p^{1,3} = 0.008}$ $\boldsymbol{p^{2,4} = 0.0005}$ |

**Notes.**
*Numbers on values refer to each animal group.
*$p$ values less than 0.05 are highlighted in bold.

the appearance of thymic bodies in the form of cyst-like cavities (*Postovalova et al., 2019b*). At the same time, the relationship between thymic involution and tumor progression was demonstrated (*Wang et al., 2020*; *Lagou & Karagiannis, 2023*). Thymic involution resulted in decreased naive T-cell production and a limited T-cell receptor repertoire, potentially impairing immune surveillance of neoplasia (*Schreiber et al., 2020*). Decreased peripheral immune surveillance and antitumor immunity were thought to favor the development of pretumor lesions and immune escape by newly transformed tumor cells (*Wang et al., 2020*; *Lagou & Karagiannis, 2023*). In this work during CAC, only in tolerant mice, an increase in the cortex volume fraction was observed with an increase in the thymic bodies in the form of cyst-like cavities number, which was characteristic of a chronic inflammatory process, while in the susceptible mice there was a tendency to the cortex volume fraction decrease, *i.e.,* thymic involution. Probably the morphological features of thymic reactions in mice with different resistance to hypoxia were associated with different rates of tumor progression. It is known that CAC develops as a result of epithelial dysplasia against the background of chronic inflammation with subsequent tumor formation (*Zhang et al., 2023*). Thus, in tolerant to hypoxia mice, an earlier stage of CAC was noted, since changes in the morphology of the distal colon and thymic cortex widening are more characteristic of a chronic inflammatory process, while in susceptible—of the adenocarcinoma stage.

When examining the spleen of both tolerant and susceptible mice of the experimental groups, the increase in the lymphoid follicles volume fraction was observed, but only susceptible demonstrated the lymphoid follicles germinal centers expansion. This indicates more pronounced antigenic stimulation and activation of the B-dependent zone (*Bronte & Pittet, 2013*) and was confirmed by the increase in the peripheral blood and MLNs B-lymphocytes number only in susceptible mice. Previously, we demonstrated that during

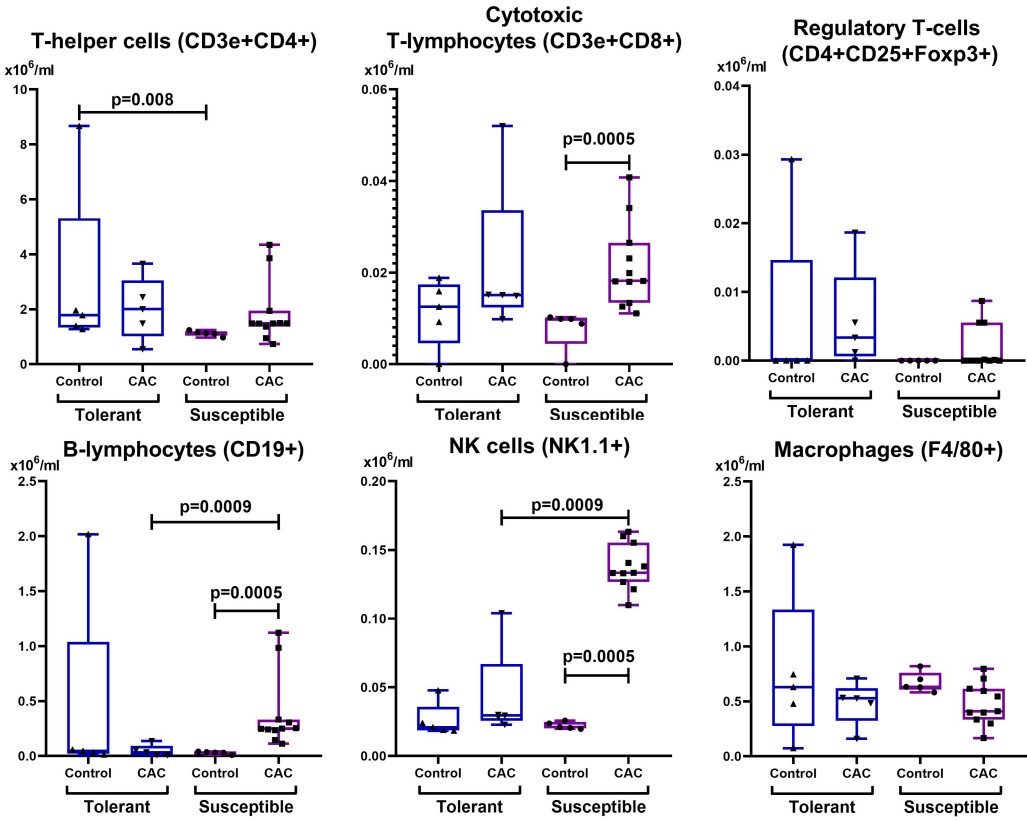

**Figure 7  Absolute MLNs cells number.** CD3e+CD4+ (T-helper cells), CD3e+CD8+ (cytotoxic T-lymphocytes), CD4+CD25+Foxp3+ (regulatory T-lymphocytes), CD19+ (B-lymphocytes), NK-1.1 (NK cells) and F4/80 (macrophages) absolute numbers in MLNs in tolerant ($n_{control} = 5$, $n_{cac} = 5$) and susceptible ($n_{control} = 5$, $n_{cac} = 11$) to hypoxia mice of the control and experimental groups.

a systemic inflammatory response induced by lipopolysaccharide, only in susceptible to hypoxia rats, predominantly activation of the humoral immunity component was observed (*Dzhalilova et al., 2019b*), which was also accompanied by an increase in the blood B-lymphocytes number and the spleen lymphoid follicles germinal centers expansion. Regional hypoxia variations involve normally germinal centers physiology, and HIF-dependent oxygen sensing regulates vital B-cells functioning. *Cho et al. (2016)* demonstrated that oxygen in lymphoid organs restriction, which can be altered in pathophysiological states, plays a modulatory role in humoral immunity. Presumably, these processes are more pronounced in susceptible to hypoxia mice.

In both tolerant and susceptible mice of experimental groups, there was an increase in the number of monocytes in the peripheral blood. It is known that monocytes, M2 macrophages and myeloid-derived suppressor cells (MDSCs) suppress the effector functions and proliferation of T-cells (*Kanterman, Sade-Feldman & Baniyash, 2012*), and also reduce the expression of activation markers on NK cells (*Wang & DuBois, 2015*). At the same time, during chronic inflammatory processes in the secondary lymphoid organs, antigens, including tumor, are presented for naive T-cells (*Lorusso & Rüegg, 2008*). The antitumor

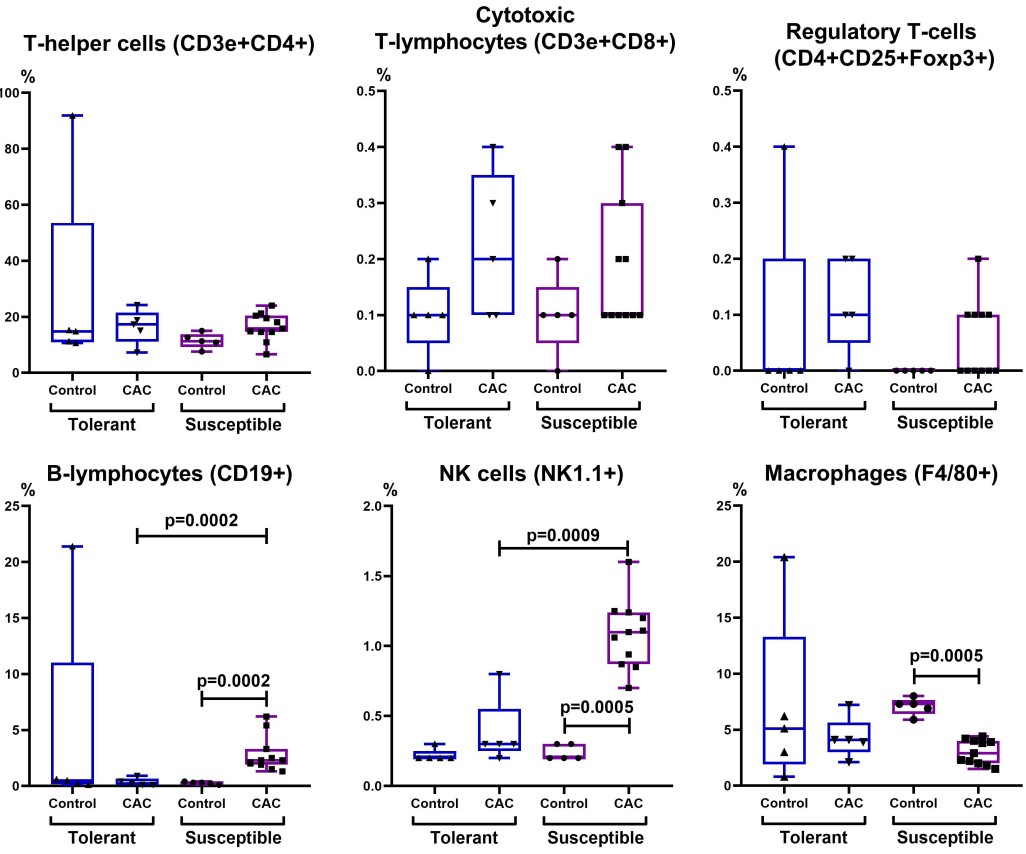

**Figure 8** **Relative MLNs cells number.** CD3e+CD4+ (T-helper cells), CD3e+CD8+ (cytotoxic T-lymphocytes), CD4+CD25+Foxp3+ (regulatory T-lymphocytes), CD19+ (B-lymphocytes), NK-1.1 (NK cells) and F4/80 (macrophages) relative numbers in MLNs in tolerant ($n_{control}$ = 5, $n_{cac}$ = 5) and susceptible ($n_{control}$ = 5, $n_{cac}$ = 11) to hypoxia mice of the control and experimental groups.

immune response was realized through the action of CD8+ cytotoxic T-lymphocytes (*Biswas & Mantovani, 2010*; *Ngiow et al., 2011*; *De Vries et al., 2016*) and upon activation of CD4+ Th1 cells (*Rao, Gharib & Han, 2019*). It is likely that the decrease in the blood CD3e+CD4+ and NK cells number of both tolerant and susceptible mice is associated with a violation of their proliferation against the background of chronic inflammation or their migration to secondary lymphoid organs, which was also consistent with an increase in the NK cells number in MLNs in susceptible to hypoxia mice. At the same time, the blood CD3e+CD8+ cytotoxic T-lymphocytes number increased, and CD4+CD25+Foxp3+ Tregs decreased. Treg cells are known to be suppressors of CD8+ cells (*Biswas & Mantovani, 2010*; *Ngiow et al., 2011*; *De Vries et al., 2016*). With a decrease in the Tregs number, it is likely that CD8+ T cells can restore their role in suppressing tumor growth (*Olguín et al., 2018*). According to literature, high density of tumor-infiltrating Tregs expressing Foxp3 correlated with poor outcome in breast (*Bates et al., 2006*), ovaries (*Curiel et al., 2004*), lungs cancer (*Petersen et al., 2006*), hepatocellular and renal cell cancer (*Gao et al., 2007*; *Li et al., 2009*), pancreatic cancer (*Hiraoka et al., 2006*), gastric cancer (*Hiraoka et al., 2006*;

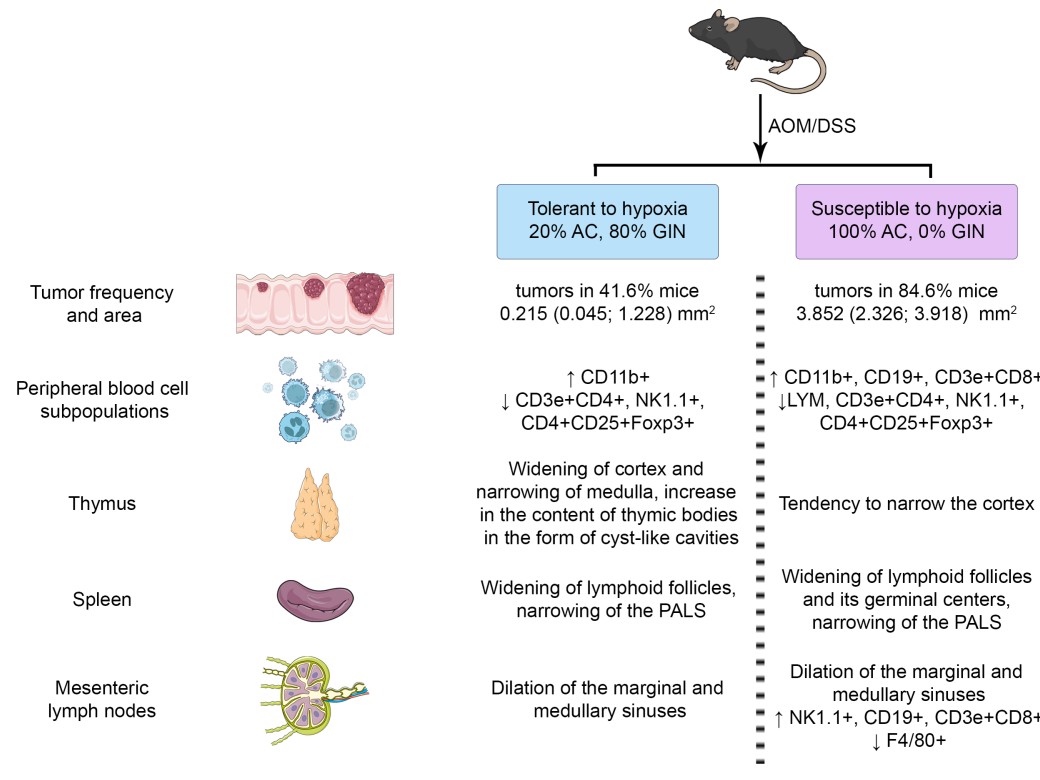

AOM/DSS

|  | Tolerant to hypoxia<br>20% AC, 80% GIN | Susceptible to hypoxia<br>100% AC, 0% GIN |
|---|---|---|
| Tumor frequency and area | tumors in 41.6% mice<br>0.215 (0.045; 1.228) mm$^2$ | tumors in 84.6% mice<br>3.852 (2.326; 3.918) mm$^2$ |
| Peripheral blood cell subpopulations | ↑ CD11b+<br>↓ CD3e+CD4+, NK1.1+,<br>CD4+CD25+Foxp3+ | ↑ CD11b+, CD19+, CD3e+CD8+<br>↓LYM, CD3e+CD4+, NK1.1+,<br>CD4+CD25+Foxp3+ |
| Thymus | Widening of cortex and narrowing of medulla, increase in the content of thymic bodies in the form of cyst-like cavities | Tendency to narrow the cortex |
| Spleen | Widening of lymphoid follicles, narrowing of the PALS | Widening of lymphoid follicles and its germinal centers, narrowing of the PALS |
| Mesenteric lymph nodes | Dilation of the marginal and medullary sinuses | Dilation of the marginal and medullary sinuses<br>↑ NK1.1+, CD19+, CD3e+CD8+<br>↓ F4/80+ |

**Figure 9** **Summary results.**

*Perrone et al., 2008*) and cervix cancer (*Jordanova et al., 2008*). However, for many years, clinical and experimental studies did not confirm this relationship in colorectal cancer, as well as in head and neck tumors (*Salama et al., 2009*; *Ladoire, Martin & Ghiringhelli, 2011*; *De Leeuw et al., 2012*; *Li et al., 2024*). One of the reasons for the patients with a high level of Treg cells favorable outcome is their ability to reduce the activity of the inflammatory response caused by the microbiota, thereby increasing the resistance of the tissue to tumor transformation (*Lam et al., 2021*). Perhaps the decrease in the blood Treg cells number of both tolerant and susceptible to hypoxia mice is associated with their migration, and the increase in the blood and in MLNs cytotoxic lymphocytes number in susceptible mice is the response for adenocarcinoma development.

The revealed morphofunctional differences in the immune system of mice with different tolerance to hypoxia may be related to different energy metabolism, which is the individual tolerance to hypoxia determinant. It was demonstrated that the number of mitochondria with more densely packed cristae and darker matrix and the number of small functionally more active mitochondria, as well as the mitochondrial enzymes concentration, are larger and higher in tolerant rats than in susceptible ones (*Pavlik et al., 2018*; *Mironova et al., 2019*). Moreover, in performed experiments of the effects of L-arginine and Nω-nitro-L-arginine administered before and after exposure to lead nitrate on animals with different tolerance to hypoxia, it was shown that a characteristic feature of animals with high tolerance to hypoxia was a faster switching of metabolic pathways involving oxygen-dependent

processes to adapt to the exposure (*Tkaczenko et al., 2024*). Hypoxia and HIF activation are known to alter immune cell metabolism. Under hypoxic conditions, increased HIF activity therefore decreases mitochondrial oxygen consumption and mitochondrial mass, and increases glycolysis, fatty acid synthesis and glutaminolysis (*Taylor & Scholz, 2022*). It is known that NF-κB directly affects the expression and content of HIF-1α protein both in hypoxia and normoxia, and in inflammatory processes, which is explained by the presence of the NF-κB binding site in the proximal part of the *HIF1A* gene promoter (*Rius et al., 2008*; *Van Uden, Kenneth & Rocha, 2008*; *Van Uden et al. 2011*). Induction of HIF-1 by NF-κB-dependent pathway activates pro-inflammatory cytokine genes (*Jantsch et al., 2011*). Hyperactivation of inflammatory molecule production leads to hemodynamic disturbances, coagulation activation with increased NO synthesis, vasoconstriction and hypoxia development, which also affects immune cells (*Cohen, 2002*). Furthermore, it is known that HIF-1α activates Th17 and CD8+ cytotoxic T lymphocytes, which stimulate macrophages to synthesize pro-inflammatory cytokines, leading to inflammation (*Barbi, Pardoll & Pan, 2013*; *Pazmandi et al., 2019*). This confirms the obtained data and suggestions about the more pronounced inflammation in susceptible to hypoxia mice of the experimental group, as only in them the relative and absolute CD3e+CD8+ lymphocytes number in the peripheral blood and the absolute number in the MLNs were increased. It is likely that baseline differences in the HIF expression, the level of which was higher in susceptible to hypoxia animals (*Kirova, Germanova & Lukyanova, 2013*; *Jain et al., 2013*; *Jain et al., 2014*; *Dzhalilova et al., 2019a*), largely determine the functioning of immune cells in CAC in these animals. During CAC we previously demonstrated that changes in the genes expression regulating hypoxia response, inflammation, cell cycle, apoptosis, and epithelial barrier function in tumors and the peritumoral area, as the number of CD3+CD8+ cytotoxic T-lymphocytes and vimentin+ cells, depend on the initial hypoxia tolerance. Higher expression of *Hif3a, Vegf, Tnfa, Il10, Tgfb, Cmet, Egf, Egfr, Egfr, Bax, Muc1*, and *Cldn7* was observed in the tumors of susceptible to hypoxia animals compared to tolerant mice, while only *Egf* was higher in the peritumoral area. Moreover, the content of F4/80+ macrophages, B-lymphocytes, T-helpers and NK cells in tumors did not differ between animals with different tolerance to hypoxia, but the number of cytotoxic T lymphocytes and vimentin+ cells was higher in hypoxia susceptible mice (*Dzhalilova et al., 2024a*). A more pronounced immune response to the CAC development in susceptible to hypoxia mice may be associated with less effective defense mechanisms, including antioxidant defense enzymes, heat shock proteins (*Jain et al., 2013*; *Jain et al., 2014*; *Padhy et al., 2013*), and immune response against the chronic inflammation background.

## CONCLUSION

It was demonstrated that on the 141st day after the azoxymethane administration and three cycles of dextran sulfate sodium consumption, in susceptible to hypoxia mice, a higher incidence of tumor development was observed in comparison to tolerant mice. In addition, in susceptible to hypoxia mice, 100% of detected tumors were represented by adenocarcinomas, while in tolerant—20% adenocarcinomas and 80% glandular

intraepithelial neoplasia. Immunohistochemical study also revealed more pronounced Ki-67+ staining in tumors of susceptible mice, indicating high rate of tumor progression compared to tolerant mice. In susceptible to hypoxia mice, the process of tumor initiation and progression against the background of chronic inflammation occurs faster and is accompanied by more pronounced changes in the subpopulation composition of blood cells and MLNs, as well as by expansion of the spleen lymphoid follicles germinal centers, while in tolerant mice, the thymus cortex expansion and the medulla narrowing was observed relative to the control group. Morphofunctional differences in the immune system reactions indicate a more pronounced immune response to the CAC development in susceptible to hypoxia mice, which should be taken into account in further CAC studies. Limitations of our study are connected with lack of comprehensive proliferation analysis beyond Ki-67 (*e.g.*, additional markers like PCNA or functional proliferation assays). Such data can contribute to the future perspectives of the study.

### Funding
The work was supported by the Russian Science Foundation (grant No. 23-25-00294 'Individual tolerant to hypoxia and molecular-biological features of tumor growth initiation in an experimental model of colorectal cancer'). The funders had no role in study design, data collection and analysis, decision to publish, or preparation of the manuscript.

### Grant Disclosures
The following grant information was disclosed by the authors:
Russian Science Foundation: 23-25-00294.

### Competing Interests
The authors declare there are no competing interests.

### Author Contributions
- Dzhuliia Dzhalilova conceived and designed the experiments, performed the experiments, analyzed the data, prepared figures and/or tables, authored or reviewed drafts of the article, and approved the final draft.
- Maria Silina performed the experiments, analyzed the data, prepared figures and/or tables, authored or reviewed drafts of the article, and approved the final draft.
- Anna Kosyreva performed the experiments, analyzed the data, authored or reviewed drafts of the article, and approved the final draft.
- Nikolai Fokichev analyzed the data, prepared figures and/or tables, authored or reviewed drafts of the article, and approved the final draft.
- Olga Makarova conceived and designed the experiments, analyzed the data, authored or reviewed drafts of the article, and approved the final draft.

### Animal Ethics
The following information was supplied relating to ethical approvals (i.e., approving body and any reference numbers):

The Bioethical Committee at Avtsyn Research Institute of Human Morphology provided full approval for this research (Protocol No. 36/12, March 28, 2022).

## Data Availability

The flow cytometry data is available at Flow Repository: FR-FCM-Z7HS.

The raw measurements are available in the Supplemental File.

## Supplemental Information

Supplemental information for this article can be found online at http://dx.doi.org/10.7717/peerj.19024#supplemental-information.

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
