# Peer review of "Morphofunctional changes in the immune system in colitis-associated colorectal cancer in tolerant and susceptible to hypoxia mice"

_PeerJ, doi:10.7717/peerj.19024_

## Round 0.1 · original submission · Major Revisions

Dear authors, thank you for your patience.

Thank you for resubmitting your manuscript, which investigates the influence of hypoxia tolerance on immune responses during colitis-associated colorectal cancer (CAC) in a mouse model.

I think that it comes at no surprise that an in vivo work on hipoxia can generate a lot of curiosity as well as questions. So, while the study demonstrates some promising observations in immune morphology and CAC development, the reviewers have identified several major areas for improvement that need to be addressed to strengthen the manuscript and support its publication. Please, refer to the reviewers' comments, particularly in experimental design and mechanistic depth which are essential to bring your findings in line with the journal's standards.

Below, I try to outline the key concerns:

- although the study hints at the role of hypoxia-inducible factor (HIF) in immune modulation and CAC, there is no direct discussion or mechanistic evidence involving HIF pathways in this study. It is advised that you either clarify the relevance of HIF to your work more explicitly or conduct supplementary analyses on HIF's role in CAC progression under hypoxic conditions.
- While your work captures morphological immune differences between hypoxia-tolerant and susceptible mice, reviewers emphasize the need for deeper mechanistic insights. The addition of data on HIF-1α target gene expression (e.g., VEGF, GLUT1, CXCR4) would be highly beneficial, offering clarity on the molecular pathways through which hypoxia and HIF regulate immune responses in CAC. Such data could substantiate your findings and create a robust foundation for future clinical implications.
- Clarifying the clinical implications of hypoxia tolerance in CAC and establishing the broader relevance of your model could strengthen the study’s impact. Furthermore, given that the research is conducted solely in a mouse model, reviewers recommend moderating claims about the clinical applicability of your findings until additional translational evidence is available.
- Improving data presentation is essential for clear communication of your findings. This includes more precise figure layouts, consistent use of scale bars, and specifying gating strategies for flow cytometry data. A clear, representative figure layout, with additional morphological images as needed, would significantly enhance the manuscript’s clarity.
- there are some concerns in the methodological rigor, particularly regarding the assessment of hypoxia tolerance and ethical considerations in animal housing. Ensuring that hypoxia tolerance is accurately measured and that institutional approval for co-housing 10 mice per cage is clearly indicated will address these concerns. Clarifying specific techniques (e.g., for cell type identification) and ensuring clear figure legends with biological sample numbers will also add robustness to the study.

Conducting an in vivo study like yours is complex and demanding, requiring high standards in experimental design and clear translational relevance. Addressing these major revisions can offer you an opportunity to enhance the scientific value and clinical impact of your important work, which reviewers see as highly promising but currently lacking sufficient mechanistic and translational detail for publication.
Please consider these recommendations carefully as you prepare your revised manuscript. I look forward to reviewing your work again.

·

Basic reporting

no comment

Experimental design

1.The background section of the article mentions that hypoxia-inducible factor (HIF) is related to the immune process and the occurrence of CAC, but there is no content related to HIF in this study. Therefore, it is recommended that the authors clearly indicate the relevance of HIF to the research theme in the article, or supplement the role of HIF in the occurrence of CAC in subsequent studies.
2. This paper establishes a CAC model in hypoxia-tolerant and susceptible mice and explores the differences in immune system morphology and function, which has a certain degree of innovation. However, the depth of the research is not sufficient. Although the article mentions the differences in immune system response in morphology and function, there is no in-depth mechanistic study behind the differences between the two groups. It is recommended that the authors conduct thorough animal and mechanistic studies to further clarify the relationship between hypoxia, immune regulation, and cancer.

Validity of the findings

1. This study selects hypoxia as a research variable, which is a promising research direction. However, the article fails to deeply explain the specific connection between hypoxia and CAC from a clinical perspective. Therefore, the research intention appears unclear, lacking direct guiding significance for clinical practice.
2.The conclusions of the article are somewhat generalized because the experimental results are based solely on a mouse model and may not necessarily apply to humans.

Reviewer 2 ·

Basic reporting

This study investigates the role of hypoxia tolerance in modulating immune responses during colitis-associated colorectal cancer (CAC) in mice, focusing on the effects of hypoxia-inducible factor (HIF) activity. Hypoxia tolerance was assessed in animals, categorizing them as hypoxia-tolerant or hypoxia-susceptible, followed by CAC induction through azoxymethane and dextran sulfate sodium. Morphological analysis revealed that tumor size was larger in hypoxia-susceptible mice, alongside significant changes in immune cell populations, including increased CD19+ B cells, CD8+ T cells, and NK cells, particularly in susceptible animals. Additionally, lymphoid tissue architecture differed between groups, with alterations in the thymus and spleen. This study highlights that hypoxia susceptibility is associated with an amplified immune response in CAC, suggesting that hypoxia tolerance may be an important consideration in designing effective immunotherapies for CAC.

While the study describes morphofunctional differences and changes in immune cell populations, it lacks in-depth mechanistic insights into how hypoxia and HIF regulation specifically influence the immune responses and CAC development. Including potential molecular pathways could strengthen the discussion.

Experimental design

1- The study describes groups categorized as "tolerant to hypoxia," "normal," and "susceptible to hypoxia," but it may be beneficial to clarify how the "normal" group differs from the others. A proper control group is essential to establish baseline measurements and ensure the validity of the results.

2- The method used to assess hypoxia tolerance (gasping time measurement in a decompression chamber) may not fully capture the complex physiological adaptations to hypoxia. Providing additional measures or validation of hypoxia tolerance could improve robustness.

3- The study lacks data on hypoxia-inducible factor 1-alpha (HIF-1α) target gene expression, which limits insights into the molecular mechanisms underlying hypoxia’s role in CAC progression. HIF-1α regulates critical genes like VEGF, GLUT1, and CXCR4, which are involved in angiogenesis, metabolic adaptation, and immune cell trafficking—processes central to tumor growth and immune modulation in hypoxic conditions. Without examining these responsive genes, the study misses a potential link between observed immune cell changes and specific hypoxia-driven pathways, reducing the depth of mechanistic understanding. Future investigations measuring HIF-1α-regulated genes could clarify how hypoxia tolerance impacts immune response and tumor behavior in CAC, providing a stronger foundation for therapeutic strategies.

Validity of the findings

To confirm hypoxia tolerance or susceptibility, I recommend assessing HIF-1α-responsive genes, as their expression provides a direct measure of cellular adaptation to hypoxia. Key genes regulated by HIF-1α—such as VEGF for angiogenesis, GLUT1 for glucose uptake, and CXCR4 for immune cell migration—offer valuable insights into the physiological responses underpinning hypoxia tolerance. By evaluating the expression of these genes, the study could better determine whether the observed immune and morphological differences in CAC development correlate with effective HIF-1α-mediated hypoxic adaptation. This approach would strengthen the characterization of hypoxia tolerance and its role in modulating tumor progression and immune responses.

·

Basic reporting

The manuscript employs a mouse model to investigate the development of CAC under hypoxic conditions, which is interesting and novel. However, several critical issues need to be addressed. Firstly, the authors did not provide a thorough rationale linking the observed morphological changes in the thymus, the immune cell populations in the MLN and peripheral blood to the development of CAC. I recommend that the authors emphasize the changes that driven by hypoxia in the CAC. The authors could consider including additional morphological analyses to characterize the CAC for the four experimental groups (e.g, Ki-67 and aSMA immunohistochemistry for CAC) or perform immune cell analyses by flow cytometry associated with CAC.
The second essential item is the English language in the manuscript should be improved for greater precision, conciseness, and specificity to better convey the main points and help readers grasp the findings more clearly. For instance, the background and discussion sections are overly lengthy, and the authors frequently use terms such as 'organisms,' 'animals,' and 'immune system reactions,' which may confuse readers (Lines 131, 136, and 137, respectively). All subtitles should be rephrased to provide brief and clear summaries of the findings rather than fragmented sentences.

Experimental design

The mice are co-housed at a density of 10 males per cage, which may raise concerns of animal welfare. If this arrangement is approved by the institution, the authors should clearly state it in the manuscript. In the methods section, it is stated that the endpoint was when the tumor size reached 2 cm. How was this measurement done in vitro? And in method section, the term 'elimination' requires clarification. The physiological solution used for injection should be specified. Some gating methods for immune cells need corrections, e.g, CD4+ T cells are supposed to be gated as TCRb+CD4+, monocytes are gated as TCRb-CD11b+Ly6Chigh. Including a representative gating method in the figures would help convey the data more convincingly. Also, it would be helpful to specify the tools used for flow cytometry analysis (e.g., FlowJo). Another crucial point is that bar graphs with scattered points would be a more straightforward and effective way to present flow cytometry data with p-values. The figure layout is confusing. For instance, in Figure 1B and C, the figures should show the representative images for each group other than different stages of the colitis as the analysis is intended to compare the severity of colitis and the progression of CAC among groups. Supplementing Figure 1 with additional macroscopic images of the colon could make the findings more convincing. The scale bars in the figures should be consistent. In Figure 2, the ‘boundaries between the cortex and medulla’ is not labeled, which makes it hard to identify the differences among the groups. Also, the 4 groups should be clearly stated in the figures for clarity. Figure legends should also include how many biological samples are used for each analysis.

Validity of the findings

Some findings in the discussion sessions are overstated. For example, the authors mentioned immune cell migration from peripheral blood and MLN to the colon in the discussion, though if there is no data to support this hypothesis, the discussion should be simplified.

Additional comments

In Lines 330 and 337, the authors mentioned the observation of macrophages and plasma cells in the histological images. How were these cell types identified? This identification method should be described in the manuscript. Techniques such as F4/80 staining for macrophages could be used to confirm their presence in the tissue sections.

---

## Round 0.2 · Minor Revisions

Dear authors, thank you for your resubmission. We appreciate the effort you and your team have put into this work. After thorough evaluation I have reached a decision of minor revisions.
Key points that still need to be addressed include:
- although i do agree that redundancy should be avoided, there are details that are worthwhile be mentioned in different sections in order for the reader to recognize a flow and guiding line of the content; thus, although this is suggested by a reviewer I just advise you to carefully proofread the manuscript in order to make sure that it reflects the message intended.
- That said, I recommend improving clarity in the Results section, particularly by enhancing figure legends (e.g., explicitly stating that Figure 2 represents Ki-67-positive cells and clarifying what those cells are). Providing a more comprehensive description of the immunohistochemistry (IHC) results, including detailed insights into Ki-67 findings and their interpretation, would greatly benefit the manuscript. For example. This is probably the weakest point in the results reported.
- Address the oversight regarding F4/80 staining, which is mentioned in the Methods but lacks corresponding results or images in the current manuscript (i think)
- Revise the gating strategy for CD4+ and CD8+ T lymphocytes to reflect TCRb+ or CD3e+ CD4+/CD8+ cells, as markers like CD4 and CD8 may also label innate lymphoid cells.
- If possible reduce the reliance on citations from the authors' research group, especially in cases where alternative references are available or cite also those other sources
- Careful on how you phrase statements: (e.g., Lines 452 and 456) that compare the current findings to previous work to emphasize the novelty and unique contributions of this study.

Additionally, I advise to:
- Ensure all figure legends are descriptive and provide sufficient detail for readers to understand what is being shown without referencing the main text.

Overall, we appreciate the improvements made to the manuscript and look forward to seeing your revisions.

Reviewer 2 ·

Basic reporting

I think the authors have performed fundamental changes and improved the manuscript. I would accept for publication

Experimental design

I am satisfied with the experimental design in the revised version of the manuscript

Validity of the findings

I think the findings are valid and well proven experimentally

Additional comments

No more comments

·

Basic reporting

The background and discussion sections are lengthy, meanwhile, the 'Tumor Immunohistochemical Study' section is quite brief. The authors could shorten the background and discussion by eliminating redundant details or transferring some of the content to the Results section. The authors should extend the 'Tumor Immunohistochemical Study' by including detailed descriptions of the Ki-67 IHC findings and providing insights into how these findings. Have the authors conducted quantification Ki-67? F4/80 staining is mentioned in the methods, but corresponding results or images are missing. Please correct the gating of CD4 and CD8 T lymphocytes to TCRb+ or CD3e+ CD4+/CD8+ cells, as some innate lymphoid cells also express the markers though they lack lineage markers TCRb or CD3e.

Experimental design

nothing in particular

Validity of the findings

Several papers from the same research groups (as the authors) have been cited and mentioned repeatedly. To avoid potential bias, it would be better to reduce the number of citations from these groups and include a broader range of references from other sources (e.g. Line 111) In the line 452 and 456, the authors stated that 'the results was consistent with our previously published papers'. This statement could be rephrased to enhance the novelty of the current study, potentially by highlighting how the findings build upon or extend their previous work.

Additional comments

none

---

## Round 0.3 · accepted · Accept

Dear authors,

Thank you for your work. I am approving your manuscript for publication.
After highlighting the Ki-67 situation in my last decision I took some time analyzing the pros and cons of keeping it. It is after all your work, and although Ki-67 might not be entirely redundant, particularly when working with cancer tissues, its necessity depends on the specific goals of the study.
Ki-67 does confirm the cancer state and your results do mention that Ki-67+ staining was more pronounced in tumors of hypoxia-susceptible mice (which could be a question other peers could also raise). So this suggests that tumor proliferation differs based on hypoxia tolerance, which could add value to your findings. And since reviewers specifically pointed out its need, removing it entirely might raise concerns about omission rather than addressing the weakness they highlighted. So, i would suggest keeping it in the discussion and methods, but maybe move the results to supplementary, as your primary objective is to assess the immune response to CAC, not the proliferative activity of tumor cells. Since Ki-67 mainly reflects tumor cell proliferation rather than immune response, it may not be central to your conclusions. And since you do not include a comprehensive proliferation analysis beyond Ki-67 (e.g., additional markers like PCNA or functional proliferation assays), it might be viewed as an incomplete proliferation assessment. So, the idea is that if you cannot expand on the Ki-67 data in a meaningful way, it might be better to move it to supplementary materials and "briefly" mention it in the main text rather than keeping it as a central result. And maybe consider a brief additional statement in the results acknowledging the limitations of proliferation assessment if needed. Just a small "future perspectives" like statement at the end.

The revised version submitted is accepted. The work is generally scientifically sound and meets the standard for publication. Please, decide on this minor aspect during the proofreading stage.